# Metaphase chromosome structure is dynamically maintained by condensin I-directed DNA (de)catenation

**Ewa Piskadlo, Alexandra Tavares, Raquel A Oliveira***

Instituto Gulbenkian de Ciência, Oeiras, Portugal

**Abstract** Mitotic chromosome assembly remains a big mystery in biology. Condensin complexes are pivotal for chromosome architecture yet how they shape mitotic chromatin remains unknown. Using acute inactivation approaches and live-cell imaging in *Drosophila* embryos, we dissect the role of condensin I in the maintenance of mitotic chromosome structure with unprecedented temporal resolution. Removal of condensin I from pre-established chromosomes results in rapid disassembly of centromeric regions while most chromatin mass undergoes hyper-compaction. This is accompanied by drastic changes in the degree of sister chromatid intertwines. While wild-type metaphase chromosomes display residual levels of catenations, upon timely removal of condensin I, chromosomes present high levels of *de novo* Topoisomerase II (TopoII)-dependent re-entanglements, and complete failure in chromosome segregation. TopoII is thus capable of re-intertwining previously separated DNA molecules and condensin I continuously required to counteract this erroneous activity. We propose that maintenance of chromosome resolution is a highly dynamic bidirectional process.

*For correspondence: rcoliveira@ igc.gulbenkian.pt

**Competing interests:** The authors declare that no competing interests exist.

## Introduction

Mitotic chromosome assembly, although poorly understood at the molecular level (*Piskadlo and Oliveira, 2016*), fulfils three major tasks essential for faithful chromosome segregation: First, it ensures chromosome compaction making cell division feasible within the cell space. Secondly, it provides chromosomes with the right mechanical properties (e.g. bendiness and rigidity) to facilitate their drastic movements during mitosis. Lastly, it ensures the resolution of the topological constrains that exist between the two sister DNA molecules, as well as between neighbouring chromosomes (chromosome individualization), a key requisite for efficient chromosome partitioning. At the heart of these structural changes are the condensin complexes. Condensin complexes, one of the most abundant non-histone complexes on mitotic chromosomes (*Cuylen and Haering, 2011*; *Hirano et al., 1997*; *Ono et al., 2003*), are composed of two structural maintenance of chromosome (SMC) proteins (SMC2 and SMC4) bridged by a kleisin subunit (Barren/Cap-H for condensin I and Barren2/ Cap-H2 for condensin II)(*Cuylen and Haering, 2011*; *Hirano et al., 1997*; *Ono et al., 2003*). Despite extensive research over the last years, how condensins contribute to chromosome morphology is not completely understood. Biochemical and phenotypic analysis of condensin depletion suggest several possible activities for these complexes, including the resolution of DNA entanglements (*Gerlich et al., 2006*; *Hagstrom et al., 2002*; *Hirano, 2006*; *Hudson et al., 2003*; *Oliveira et al., 2005*; *Ribeiro et al., 2009*; *Steffensen et al., 2001*) and structural integrity by conferring chromosome rigidity (*Gerlich et al., 2006*; *Houlard et al., 2015*; *Oliveira et al., 2005*; *Ribeiro et al., 2009*). Whether or not these complexes also promote chromatin compaction remains controversial (*Hagstrom et al., 2002*; *Hirano, 2006*; *Hirano et al., 1997*; *Hudson et al., 2003*; *Kimura and Hirano, 1997*; *Lavoie et al., 2002*; *Oliveira et al., 2005*; *Steffensen et al., 2001*). The multiple

**eLife digest** Living cells can contain huge amounts of genetic information encoded in long strands of DNA. In total several metres of DNA are packed into a small space inside each human cell and these strands can easily become entangled and knotted. When a cell divides to produce new cells the DNA is duplicated and the two copies must be reliably separated, meaning all the knots must be undone. If the DNA strands are not properly separated it can cause extensive damage to genes when the cell tries to divide.

Enzymes called topoisomerases work to undo the tangles in DNA allowing it to be divided into two cells. A large protein complex called "condensin I" plays also an important part in organising DNA, and it has also been implicated in helping to resolve knots in the DNA. However, it was not known how condensin I contributes to the successful separation of DNA into new cells, or when in the course of a cell dividing the knots finally get untangled.

Cell division is similar in humans and the fruit fly *Drosophila melanogaster*, and so the fly is often used as a simple way to study this process in the laboratory. Now, Piskadlo et al. have examined the role of condensin I in dividing fruit fly cells by using recently developed techniques that rapidly shut down key molecular machineries while cells divide. The results show that condensin I and an enzyme called Topoisomerase II work together to separate entangled DNA. Topisomerase II can both entangle and disentangle DNA strands and it is condensin I that guides this process to ensure that ultimately all the knots are removed.

These findings show that successful cell division requires constant attention from condensin I to make sure Topoisomerase II aids cell division, rather than making the DNA more tangled. Overall this requires more active and constant work to disentangle DNA than expected, and further work is now needed to explain why. Understanding how cells avoid DNA damage during division clarifies why errors in this process cause diseases. For example, changes to condensin I are common in certain cancers and can also lead to disrupted brain development (e.g. microcephaly).

phenotypes observed on mitotic chromosomes upon depletion of condensin complexes raise the possibility that these complexes may have distinct roles at different times of mitosis. Additionally, several lines of evidence support that these complexes also influence interphase chromosome structure (*Cobbe et al., 2006*; *Hartl et al., 2008*). Thus, it is difficult, if not impossible to interpret the results when condensins are depleted prior to mitotic entry using conventional depletion approaches. To circumvent this limitation, we adopt a 'reverse and acute' approach to dissect the role of condensin I in the maintenance of chromosome organization. We find that inactivation of one condensin I specifically during metaphase leads to over-compaction at the majority of chromosomal regions. We further demonstrate that upon condensin I cleavage previously separated sister DNA molecules undergo topoisomerase II-dependent re-intertwining and complete failure in chromosome segregation.

## Results

### A TEV-protease mediated system to inactivate condensin I in *Drosophila melanogaster*

To study the role of condensin complexes in the maintenance of chromosome structure, specifically during metaphase, we developed a system to enable analysis of chromosomal structural changes upon rapid and temporally controlled inactivation of condensin in *Drosophila melanogaster*. Our analysis focused on condensin I complex as prior studies reveal a minor role for condensin II in mitotic chromosome organization in *Drosophila* (*Hartl et al., 2008*; *Herzog et al., 2013*; *Savvidou et al., 2005*). We developed a fast inactivation system to disrupt condensin I in the living fly (*Figure 1* and *Figure 1—figure supplement 1*), following a similar strategy previously used for the structurally related complex cohesin (*Oliveira et al., 2010*; *Pauli et al., 2008*; *Uhlmann et al., 2000*). This system is based on the use of an exogenous protease (Tobacco Etch Virus, TEV) to cleave an engineered protein of interest that contains TEV-cleavage sites and allows specific, rapid

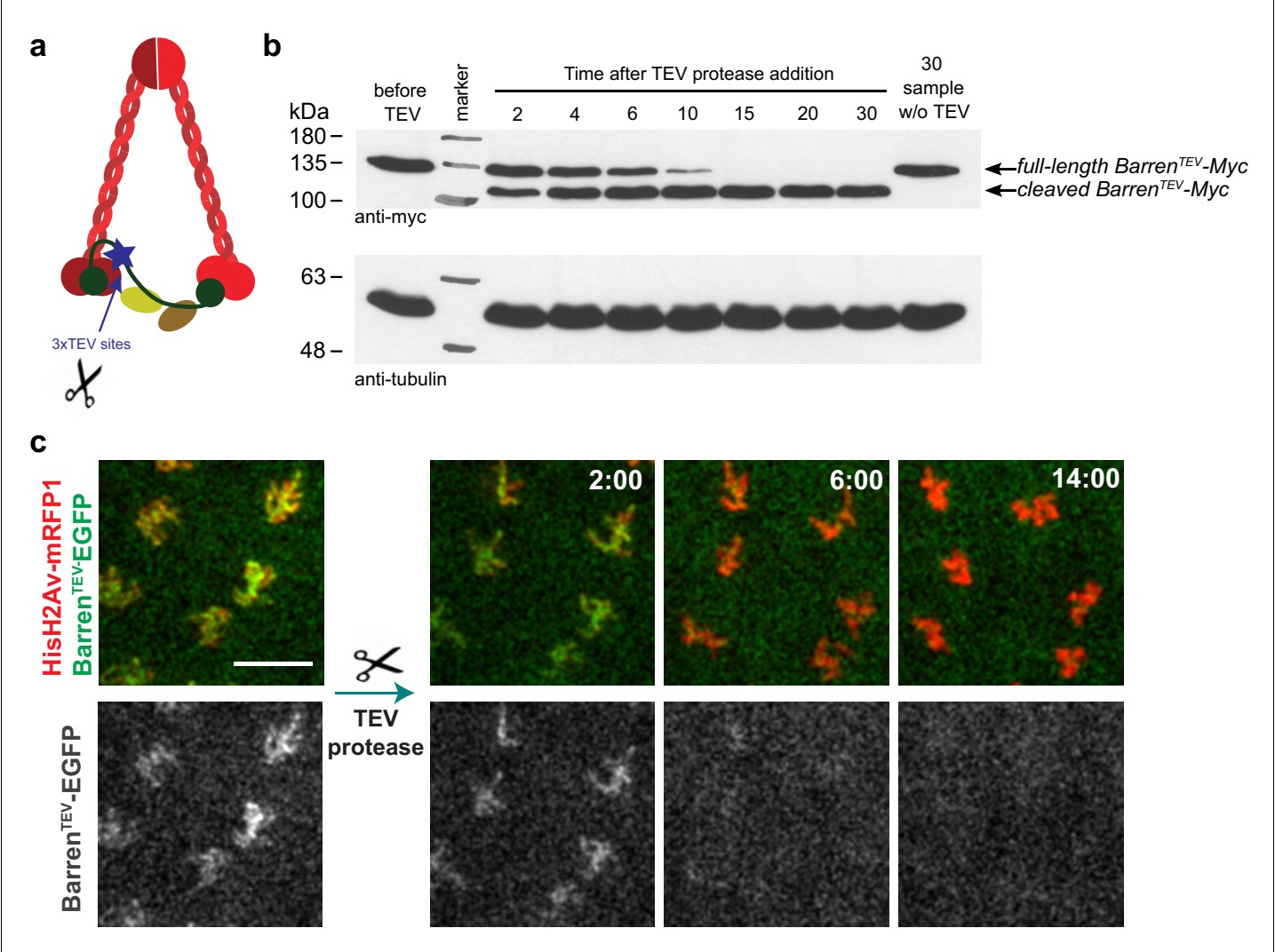

**Figure 1.** TEV-mediated cleavage of Barren disrupts condensin I function within a few minutes. (a) Schematic representation of condensin complex indicating the position of the 3xTEV cleavage sites in the kleisin subunit Barren (aa175). (b) In vitro cleavage of Barren[TEV]-myc. Extracts were prepared from ovaries of flies expressing solely TEV-cleavable Barren and incubated with TEV protease for the indicated time points (periods of time). The presence of full-length and cleaved Barren was monitored by western blot using myc antibodies. Tubulin was used as loading control. (c) Early embryos (0–30 min old) expressing HisH2AvD-mRFP1 (red) were injected with mRNA coding for Barren[TEV]-EGFP (green). Embryos were aged for 1 hr-1hr 30m to allow for protein expression. Embryos were injected with 12 mg/ml UbcH10[C114S] protein to arrest in metaphase and subsequently with TEV-protease; images depict the same region before and after TEV injection; times (minutes:seconds) are relative to the time of injection; scale bar is 10 μm.

The following figure supplement is available for figure 1:

**Figure supplement 1.** A TEV-cleavable system to destroy condensin I.

and efficient protein inactivation in a tissue- and/or time-dependent manner (*Figure 1*, *Figure 1—figure supplement 1* and data not shown). To produce flies carrying solely TEV-sensitive condensin I complexes, we produced four versions of the kleisin subunit Barren that contain three consecutive TEV-cleavage sites at four different positions: aa175, aa389, aa437, aa600 (*Figure 1—figure supplement 1*). All versions are fully functional as they were able to complement the lethality associated with the Barren null allele *Barr[L305]* (*Bhat et al., 1996*) (*Figure 1—figure supplement 1b* and data not shown). In vitro cleavage experiments reveal that all modified proteins are efficiently cleaved by TEV protease (*Figure 1B* and *Figure 1—figure supplement 1*). The construct Barren[3xTEV175]-myc

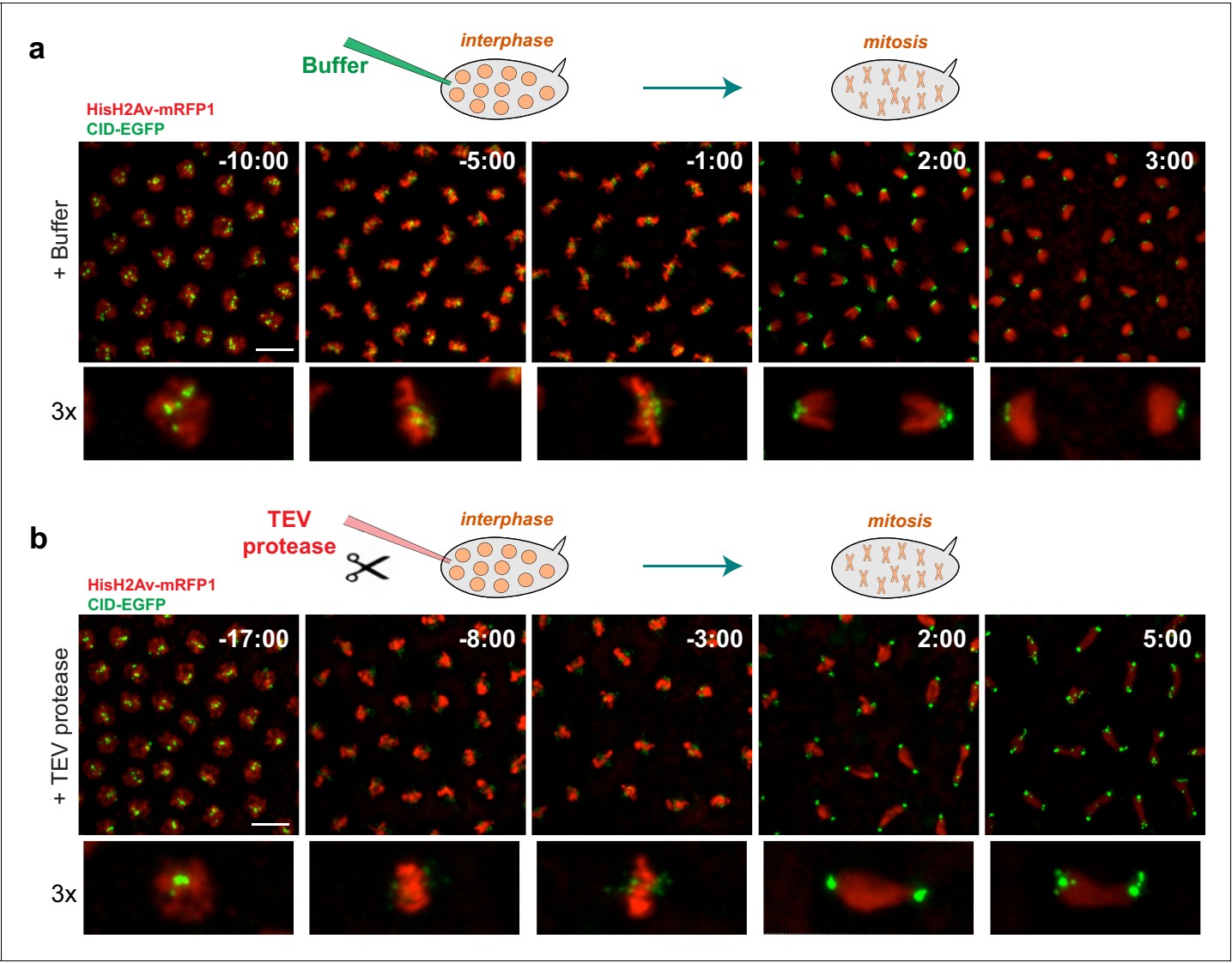

**Figure 2.** Condensin I inactivation prior to mitotic entry. Embryos surviving solely on Barren[TEV] were injected with buffer (**a**) or 13 mg/ml TEV protease (**b**) ~10–15 min before mitosis; Embryos also express His2A–mRFP1 (red) and Cid-EGFP (green); scale bars, 10 µm. Bottom rows show higher magnifications (~3x) of a single nuclear division. Times (minutes:seconds) are relative to the time of anaphase onset.

was chosen for future analysis based on the healthiness of the rescued strains (referred as Barren[TEV] hereafter).

TEV protease-mediated inactivation of condensin complexes has been previously successfully applied in yeast and mouse oocytes (*Cuylen et al., 2011*; *Houlard et al., 2015*). However, in both cases the inactivation of condensin complexes took place within over an hour after TEV protease induction. Direct injection of TEV-protease into syncytial embryos, in contrast, allowed cleavage and the removal of chromosome-associated Barren[TEV] within 8–15 min (*Figure 1b,c*), enabling the analysis of the immediate consequences upon disruption of this complex. To confirm that TEV-protease was able to inactivate condensin I efficiently within a few minutes, by cleavage of Barren[TEV], we injected TEV protease in embryos derived from females surviving solely on Barren[TEV] (ectopic expression of Barren[TEV] in a *Barr[L305]* null background). Injection of TEV-protease in early interphase embryos leads to complete failure of the subsequent mitosis (which takes place within ~15 min in these embryos). Although chromosomes were able to condense upon nuclear envelope breakdown, centromeres, monitored by the Cenp-A ortholog Cid-EGFP, display significant stretching upon

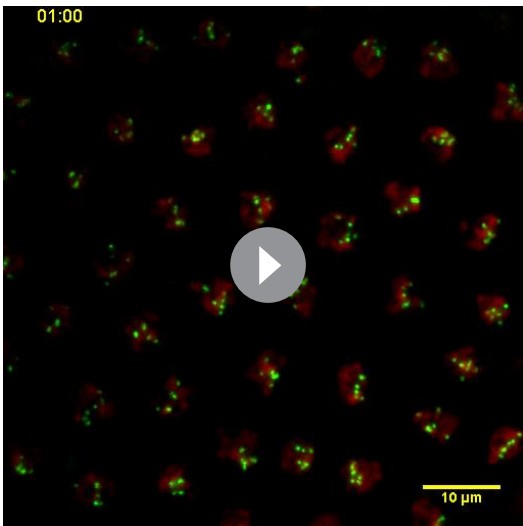

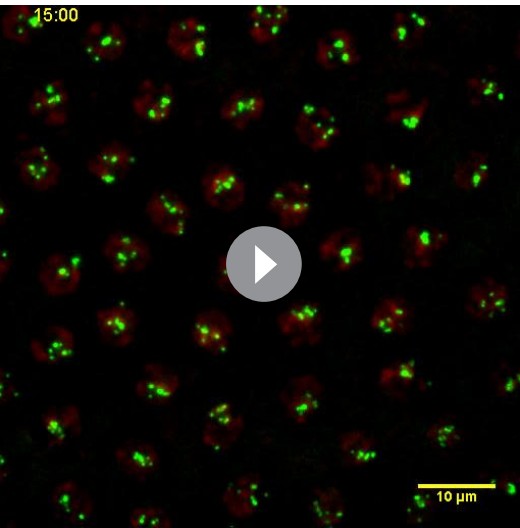

**Video 1.** Mitosis in Drosophila embryos. Embryos were injected with buffer in early interphase and monitored throughout the subsequent mitosis. Embryos express HisH2Av-mRFP1 (red) and Cid-EGFP (green). Times are relative to injection time. Scale bar is 10 um.

**Video 2.** Mitosis upon condensin I inactivation in Drosophila embryos. Embryos surviving solely on Barren[TEV] were injected with TEV protease in early interphase and monitored in the subsequent mitosis. Embryos express HisH2Av-mRFP1 (red) and Cid-EGFP (green). Times are relative to injection time. Scale bar is 10 um.

microtubule attachment (*Figure 2*, *Video 1* and *Video 2*). Moreover, resolution of sister chromatids is completely impaired, as chromatids appeared as a fused chromatin mass or display very thick bridges during the attempted anaphase (*Figure 2*, *Video 1* and *Video 2*). These results are in accordance with previous findings for condensin I depletion (*Gerlich et al., 2006*; *Hagstrom et al., 2002*; *Hirano, 2006*; *Hudson et al., 2003*; *Oliveira et al., 2005*; *Ribeiro et al., 2009*; *Steffensen et al., 2001*), which ensures the developed system is efficient at promoting rapid condensin I inactivation.

## Condensin I inactivation in metaphase leads to increased chromosome compaction

To test the role of condensin I in the maintenance of chromosome architecture, we allowed mitotic chromosomes to assemble without any perturbation on their structure and subsequently disrupted condensin I during the metaphase-arrest. Embryos were arrested in metaphase, with a functional mitotic spindle, by the use of a catalytically dead dominant-negative form of the E2 ubiquitin ligase necessary for anaphase onset (UbcH10[C114S])(*Oliveira et al., 2010*; *Rape et al., 2006*). Arrested embryos were subsequently injected with TEV protease to destroy condensin I. Given the known role of condensin I in the rigidity of pericentromeric region of chromosomes (*Gerlich et al., 2006*; *Oliveira et al., 2005*; *Ribeiro et al., 2009*), we first tested the effect of TEV protease injection at those chromosomal sites. Whereas injection of buffer causes no change in the distance between centromeres, TEV protease injection in strains containing solely TEV-cleavable Barren results in rapid separation of centromeres, that appear to stretch towards the poles, leaving behind the majority of the chromatin mass (*Figure 3*, *Video 3* and *Video 4*). These findings imply that condensin I is not only required for the establishment of a rigid structure at the pericentromeric domains prior to metaphase, but also for the maintenance of such organization.

Surprisingly, non-centromeric regions do not follow similar disorganization and in fact appeared to become more compacted. We defined chromosome compaction by degree of chromatin density, inferred from the signal of fluorescently labelled histone H2Av-mRFP1. To quantify the changes in chromosome compaction upon condensin inactivation, we used quantitative imaging analysis to monitor the mean voxel intensity, volume and surface area of each metaphase plate, over time, in

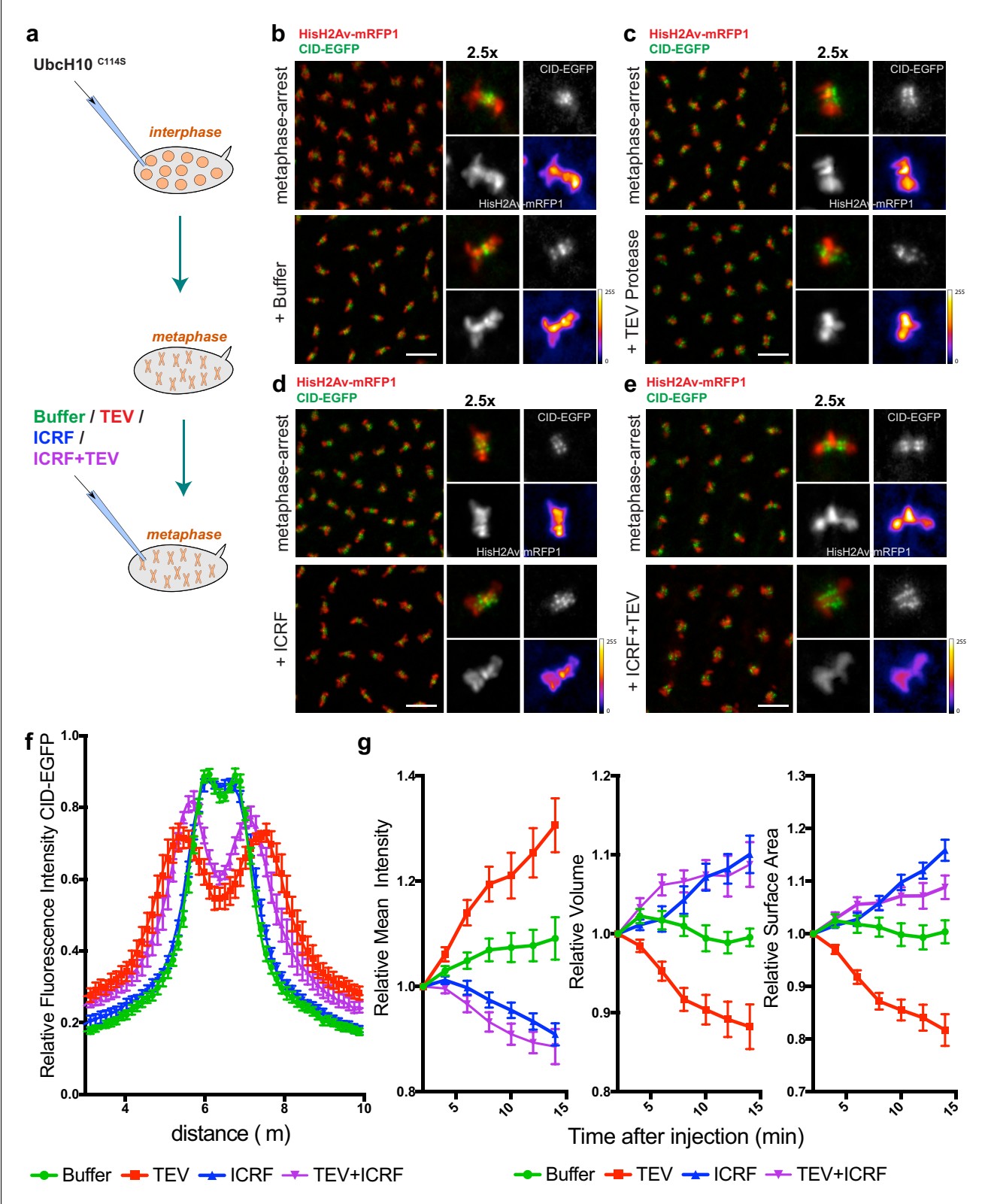

**Figure 3.** Condensin I inactivation in pre-assembled chromosomes leads to disruption of centromere structure and hyper-compaction of mitotic chromosomes. (a) Schematic representation of the experimental layout. Embryos expressing solely Barren[TEV] were injected with 12 mg/ml of a dominant-negative form of the human E2 ubiquitin-conjugating enzyme (UbcH10[C114S]) to induce a metaphase arrest. Embryos were subsequently injected with buffer (b), 13 mg/ml TEV protease (c), 280 µM ICRF (d) or a mixture containing 13 mg/ml TEV protease and 280 µM ICRF (e); Images

*Figure 3 continued on next page*

Figure 3 continued

depict embryos before the second injection and 14 min after. Embryos also express His2A–mRFP1 (red) and Cid-EGFP (green); scale bars, 10 µm. Insets show higher magnifications (2.5x) of a single metaphase. Times (minutes:seconds) are relative to the time of the second injection. (f) Quantitative analysis of centromere positioning 10 min after the second injection, as above; graph shows average ± SEM of individual embryos (n ≥ 7 embryos for each experimental condition); for each embryo, a minimum of 8 metaphases was measured; (g) quantifications of mean voxel intensity, volume and surface area of the entire metaphase plate quantified in 3D, over time, and normalized to the time of the second injection. Graphs represent the average ± SEM of individual embryos (n ≥ 10 embryos for each experimental condition); for each embryo, a minimum of 8 metaphases was quantified.

The following source data and figure supplement are available for figure 3:

**Source data 1.** Centromere displacement and chromosome compaction measurements upon condensin I and topoII inactivation.
**Figure supplement 1.** - Chromosome condensation induced by TEV-protease depends on TEV cleavage sites present in BarrenTEV.

3D (*Figure 3d*). We found that injection of TEV protease in strains surviving only on BarrenTEV leads to an overall over-compaction of the entire chromatin mass, as evidenced by an increase in the mean voxel intensity and a decrease in both the surface area and the volume of the metaphase plate (*Figure 3c,d*). Injection of the protease in strains that do not contain TEV-cleavage sites does not result in any evident change in chromosome compaction relative to controls (*Figure 3—figure supplement 1*), implying that chromosome over-compaction is specific of condensin I inactivation.

In contrast, inactivation of Topoisomerase II (TopoII) using a small molecule inhibitor (ICRF-193), leads to rapid de-compaction of chromosomes (*Figure 3d,g* and *Video 5*). TopoII has been previously implicated in chromosome compaction although its role in the process remains controversial (*Piskadlo and Oliveira, 2016*). Although we cannot exclude that chromosome decompaction may be exacerbated by the fact that ICRF-193 traps TopoII onto chromatin, our results support that TopoII may contribute to chromosome compaction in metaphase, consistent with previous observations (*Samejima et al., 2012*), possibly by promoting self-entanglements within the same chromatid (*Kawamura et al., 2010*). Importantly, combined inactivation of both TopoII and condensin I results in chromosome decompaction similar to TopoII inhibition alone (*Figure 3e,g* and *Video 6*). This finding implies that chromatin hyper-compaction observed upon loss of condensin I depends on TopoII activity.

## Condensin I inactivation results in de novo sister chromatid intertwines

The unexpected finding that condensin I inactivation promotes further chromosome compaction, together with the observation that TopoII inhibition reverts this hyper-compaction phenotype, lead us to hypothesize that the observed increase in compaction stems from re-entanglements of DNA strands, which would consequently lead to an increase in chromatin density. Enzymatically, TopoII can promote both the decatenation and the concatenation of DNA strands. Efficient chromosome segregation requires that TopoII is strongly biased towards decatenation prior to anaphase onset but it is conceivable that TopoII can additionally drive the concatenation of native metaphase chromosomes, in vivo. To test

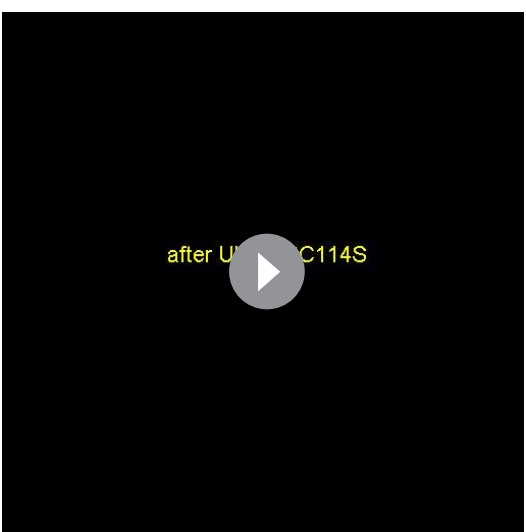

**Video 3.** Buffer injection in metaphase-arrested embryos. Embryos expressing solely BarrenTEV were injected with 12 mg/ml of a dominant-negative form of the human E2 ubiquitin-conjugating enzyme (UbcH10C114S), to induce a metaphase arrest, and subsequently injected with buffer. Embryos also express His2A–mRFP1 (red) and Cid-EGFP (green); scale bars, 10 µm. Times (minutes:seconds) are relative to the time of buffer injection.

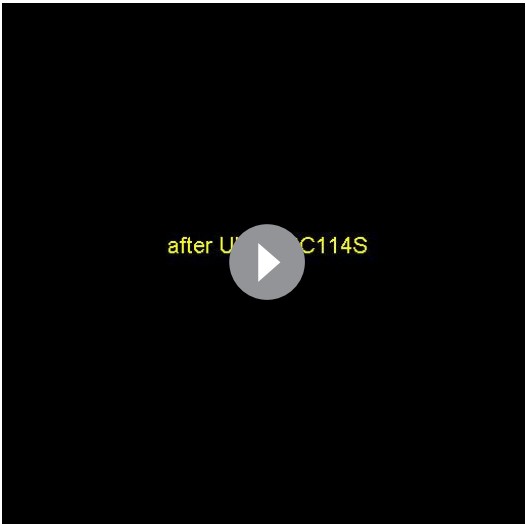

**Video 4.** Condensin I inactivation in metaphase-arrested embryos. Embryos expressing solely Barren[TEV] were injected with 12 mg/ml of a dominant-negative form of the human E2 ubiquitin-conjugating enzyme (UbcH10[C114S]), to induce a metaphase arrest, and subsequently injected with 13 mg/ml TEV protease. Embryos also express His2A–mRFP1 (red) and Cid-EGFP (green); scale bars, 10 μm. Times (minutes: seconds) are relative to the time of TEV injection.

**Video 5.** Topoisomerase II inhibition in metaphase-arrested embryos. Embryos expressing solely Barren[TEV] were injected with 12 mg/ml of a dominant-negative form of the human E2 ubiquitin-conjugating enzyme (UbcH10[C114S]), to induce a metaphase arrest, and subsequently injected with 280 μM ICRF-193. Embryos also express His2A–mRFP1 (red) and Cid-EGFP (green); scale bars, 10 μm. Times (minutes:seconds) are relative to the time of ICRF injection.

whether condensin I removal leads to re-catenation of previously separated sisters, we tested several predictions that arise from this hypothesis: First, the hyper-compaction observed in metaphase, if derived from sister-chromatid re-intertwining, should be dependent on the proximity between DNA molecules. The physical separation of sister chromatids will increase the distance between these two molecules, placing them too far apart, and consequently decreasing the likelihood of their re-entanglement, as recently proposed (*Sen et al., 2016*). Secondly, re-intertwining in late metaphase should lead to severe segregation failures. And lastly, DNA entanglements and the consequent segregation defects should depend on TopoII activity.

To evaluate the effect of sister chromatid proximity in chromosome condensation upon condensin inactivation we combined TEV-mediated cleavage of condensin I and cohesin by the use of strains carrying TEV-sensitive Rad21 (cohesin) and Barren (condensin) proteins. We took advantage of the fact that Rad21[TEV] cleavage is more efficient than Barren[TEV] (*Figure 1—figure supplement 1*), which allowed the analysis of changes in chromosome architecture upon condensin inactivation after artificial separation of sister chromatids. Upon TEV protease injection, pole-ward chromosome segregation is initiated within 2 to 5 min and with similar kinetics in both strains (*Figure 4a*).

After the initial pole-ward chromatid movement, isolated chromatids shuffle between the poles, consistent with previous observations (*Oliveira et al., 2010*). To quantify the degree of movement, we used a displacement-quantification method that infers chromosome movement by the overlap between chromosome positions on consecutive frames (*Mirkovic et al., 2015*). Cohesin cleavage alone leads to strong shuffling of isolated single chromatids, as previously described. However, concomitant inactivation of condensin and cohesin results in much slower chromatid movements, with chromatids accumulating in the middle of the segregation plane (*Figure 4b,c*). Condensin I is thus important for efficient movement of isolated chromatids. This may be due abnormal centromere/kinetochore structure and/or to a possible role for condensin in the error-correction process, as recently proposed (*Peplowska et al., 2014*; *Verzijlbergen et al., 2014*).

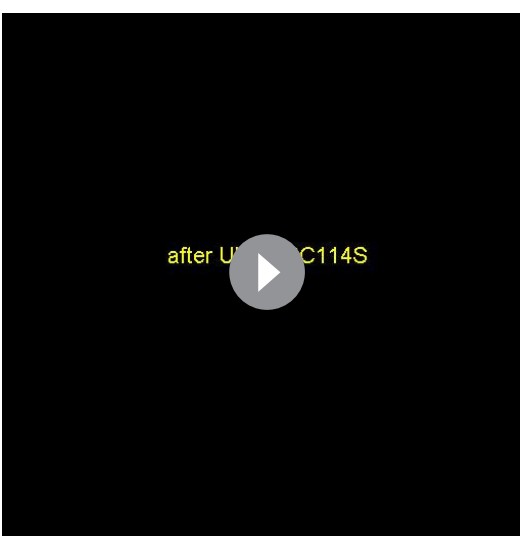

**Video 6.** Concomitant inactivation of Topoisomerase II and Condenin I in metaphase-arrested embryos. Embryos expressing solely Barren[TEV] were injected with 12 mg/ml of a dominant-negative form of the human E2 ubiquitin-conjugating enzyme (UbcH10[C114S]), to induce a metaphase arrest, and subsequently injected with a mix of 280 µM ICRF-193 and 13 mg/ml TEV protease. Embryos also express His2A–mRFP1 (red) and Cid-EGFP (green); scale bars, 10 µm. Times (minutes:seconds) are relative to the time of the second injection.

The reduced chromosome movement observed upon condensin I inactivation leads to considerable differences in chromatid positioning in both experimental set-ups. Thus, we restricted our chromosome morphology/compaction analysis to measurements of isolated single sisters, as quantifying the entire chromatin mass would include many confounding variables. Measurements of isolated single chromatids were performed at 20 min after injections and normalized to the values at 5 min after protease injection (once complete separation has occurred but no significant changes in chromosome structure was yet observed). Chromatids considerably change their shape, becoming thicker and shorter (*Figure 5b,c*, *Video 7* and *Video 8*), as previously described (*Green et al., 2012*; *Ono et al., 2003*). To directly measure the degree of compaction of these isolated sisters, we measured their mean voxel intensity. This analysis revealed that despite the significant changes in chromatid organization, there is no overall change in the mean voxel intensity of single chromatids (*Figure 5d*), indicating that shape changes are not accompanied by an overall increase in chromatin compaction. We therefore conclude that condensin I inactivation does not promote further chromosome compaction if sister chromatids are physically apart, in contrast to the effect observed in metaphase-arrested chromosomes (*Figure 3*). These results support that over-compaction observed in metaphase chromosomes may arise from sister chromatid re-intertwining, consistent with previous observations using yeast circular mini-chromosomes (*Sen et al., 2016*). It is conceivable that condensin I inactivation also promotes abnormal re-entanglement in cis (between distal regions of the same chromatid). The shape changes observed upon condensin inactivation from isolated sisters (shorter and thicker chromatids) could indeed be explained by an excess of concatenation within the same DNA molecule, leading to the shortening of the longitudinal axis. However, our compaction measurements indicate that such changes, if occur, do not lead to detectable increase in chromatin density.

Next, we sought to evaluate chromosome segregation defects, which serve as an indirect read-out for the amount of DNA catenations bridging DNA molecules. To monitor segregation defects when condensin I or TopoII are inactivated specifically in metaphase, we developed conditions in which unperturbed chromosomes would be transiently arrested in metaphase by the dominant negative UbcH10[C114S], for ~3–5 min, and subsequently injected with the respective perturbing factor in metaphase. Embryos were subsequently injected with a wild-type version of UbcH10 14 min later, which causes anaphase onset and mitotic exit in about 4–8 min (*Figure 6a*). We monitored the segregation efficiency during anaphase by quantitative analysis of the profile of Histone H2AvD-mRFP (to visualize chromatin separation) and Cid-EGFP (to infer centromere segregation) along the segregation plane (*Figure 6*). In this assay, injection of buffer causes virtually no defects in the segregation of sister chromatids (*Figure 6b*, *Figure 6—figure supplement 1* and *Video 9*).

Inactivation of TopoII under these conditions leads mostly to the appearance of fine chromatid bridges (*Figure 6c* and *Video 10*). These residual bridges are insufficient to delay centromere separation (11,01 ± 2,03 µm upon ICRF-193 treatment compared to 10,72 ± 1,69 µm in buffer-injected embryos; *Figure 6f*). The extent of chromatin bridges observed upon metaphase-specific inactivation of TopoII is considerably lower when compared to experiments where this enzyme is inhibited

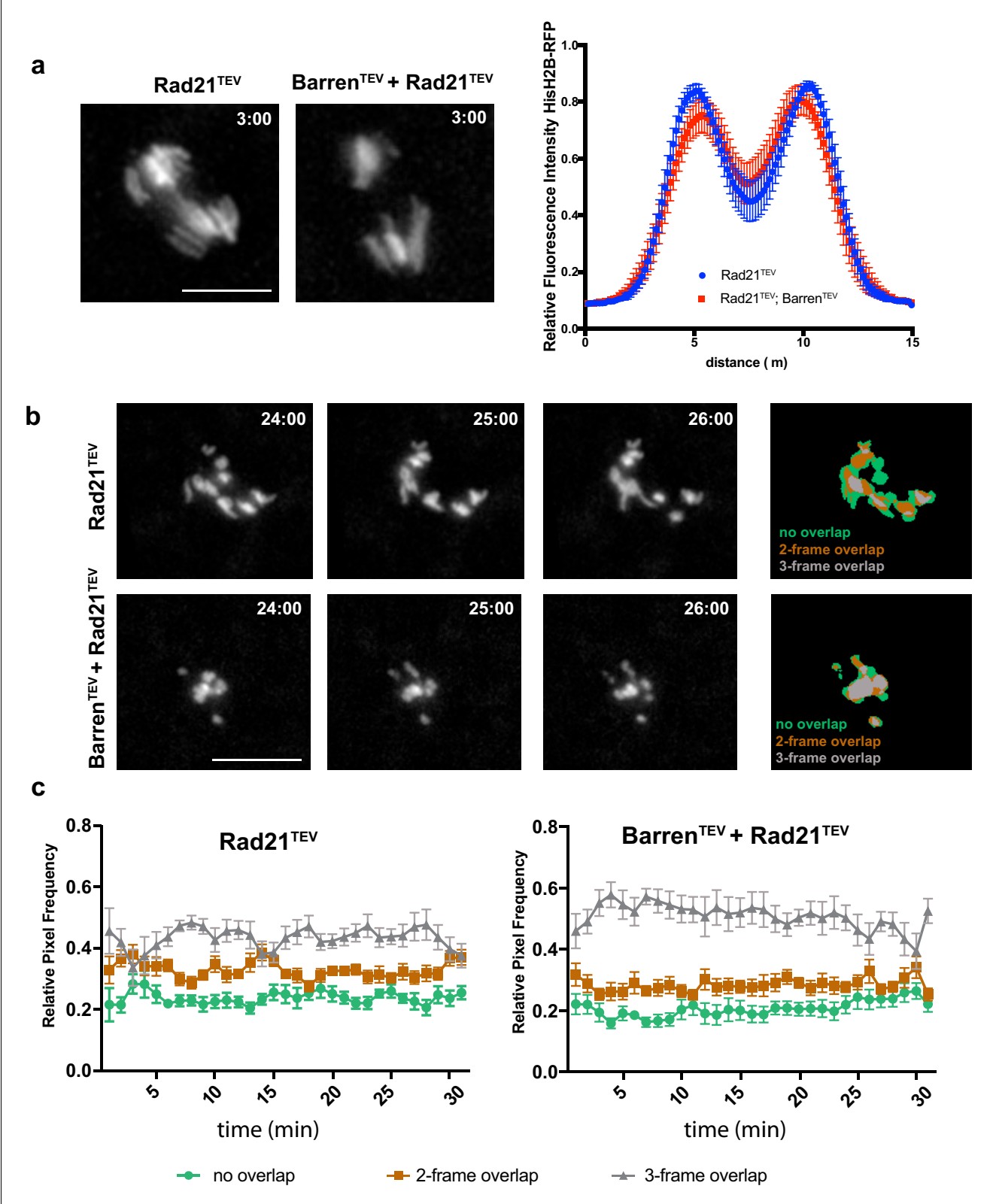

**Figure 4.** Condensin I inactivation in separated sister chromatids reduces their movement. (a) Representative images of the initial separation after TEV-mediated cleavage of Rad21[TEV] and Rad21[TEV] + Barren[TEV]. Graph plots the relative distribution of HisH2B-RFP at the maximal state of sister chromatid separation triggered by TEV-mediated cleavage of Rad21[TEV], in strains that contain solely Rad21[TEV] or both Rad21[TEV] and Barren[TEV]. A 15 μm line was used to measure plot profiles along the segregation plane, measured 3–5 min after TEV protease injection. Graphs plot the average ± SEM of

*Figure 4 continued on next page*

*Figure 4 continued*

individual embryos (n ≥ 7 embryos for each experimental condition). For each embryo, between 8 and 12 anaphases were analysed. (**b**) Example of chromosome movement analysis; left panel represents average of the binary images of three consecutive frames, used to estimate chromosome displacements: blue, non-overlapping pixels; green, two- out of three-frame overlap; grey, three-frame overlap. Scale bar is 10 μm. (**c**) Frequency of overlapping pixels to estimate chromosome displacement (as in b), over time, after TEV protease injection.
The following source data is available for figure 4:

**Source data 1.** Measurements of segregation efficiency and chromosome movement upon cohesin/condensin inactivation.

prior to mitotic entry (*Figure 6—figure supplement 1*). These findings imply that in metaphase-arrested chromosomes the amount of unresolved catenations is residual. In contrast, inactivation of condensin I during metaphase leads to complete impairment of the segregation process, as revealed by the high frequency of 'fused' chromatin masses, with the chromosomes remaining in the centre of the segregation plane, and a significant decrease in the distance between segregating centromeres (6,08 ± 0,92 μm) (*Figure 6d,f* and *Video 11*). The degree of segregation defects observed in these metaphase-inactivation experiments, is even higher than the defects observed upon condensin inactivation prior to mitotic entry (*Figure 6—figure supplement 1*). The severity of segregation impairment upon metaphase-specific condensin I inactivation indicates that in the absence of this complex previously resolved sister DNA molecules undergo re-catenation.

To directly test this hypothesis, we accessed whether or not de novo chromatin intertwines take place upon condensin inactivation, as the formation of these new links should depend on TopoII activity. If TopoII-dependent re-catenation occurs upon condensin I inactivation, one would expect that the combination of TopoII and condensin I inactivation should reduce the amount of chromatin trapped in the middle of the segregation plane. On the contrary, if segregation defects result from pre-existing catenations, combined inhibition of condensin I and TopoII should increase, or at least maintain, the density of chromosome bridges and segregation impairment.

To address this issue, we used the same experimental layout as above but induced concomitant inactivation of condensin I and TopoII during the metaphase arrest. These experiments reveal a partial rescue of the retained chromatin mass, inferred by HisH2Av-mRFP1 profile (*Figure 6e* and *Video 12*). Chromosome positioning may not be linearly correlated with the amount of linkages bridging the two sister chromatids and thus the reduction on chromosome intertwines may be even more pronounced than inferred by histone profiles. In accordance with this notion, the efficiency of chromosome separation inferred by the position of centromeres returns to levels indistinguishable from wild-type (*Figure 6e,f* and *Video 12*). Thus, concomitant inactivation of condensin I and TopoII significantly reverts the defects associated with condensin I removal. We therefore conclude that the segregation defects observed upon metaphase-specific condensin I inactivation are caused by de novo TopoII-dependent re-intertwining of previously separated sister chromatids.

## Discussion

The role of condensin complexes in chromosome compaction has been extensively debated. Here we provide evidence that temporally controlled inactivation of condensin I, specifically during metaphase, causes an increase in the overall levels of chromosome compaction in non-centromeric regions. These findings strongly argue that condensin I is required to maintain chromosomal architecture but not to sustain their compacted state. Studies using similar inactivation techniques in mouse oocytes have proposed that condensins confirm longitudinal rigidity, as chromosomes disassemble upon condensin inactivation (particularly condensin II)(*Houlard et al., 2015*). At first sight, these findings may be perceived as in sharp contrast to our current observations. It should nevertheless be noted that meiotic chromosomes are under very different force-balance than their mitotic counterparts. In particular, spindle attachment on meiotic bivalents imposes stretching along the longitudinal axis of chromosomes, similarly to what we report here for the pericentromeric region in mitotic chromosomes. Our results now demonstrate that when chromosomes are not subjected to significant additional forces, condensin I inactivation results in an overall chromatin over-compaction rather than chromosome de-condensation. This force-dependent phenotype may explain several

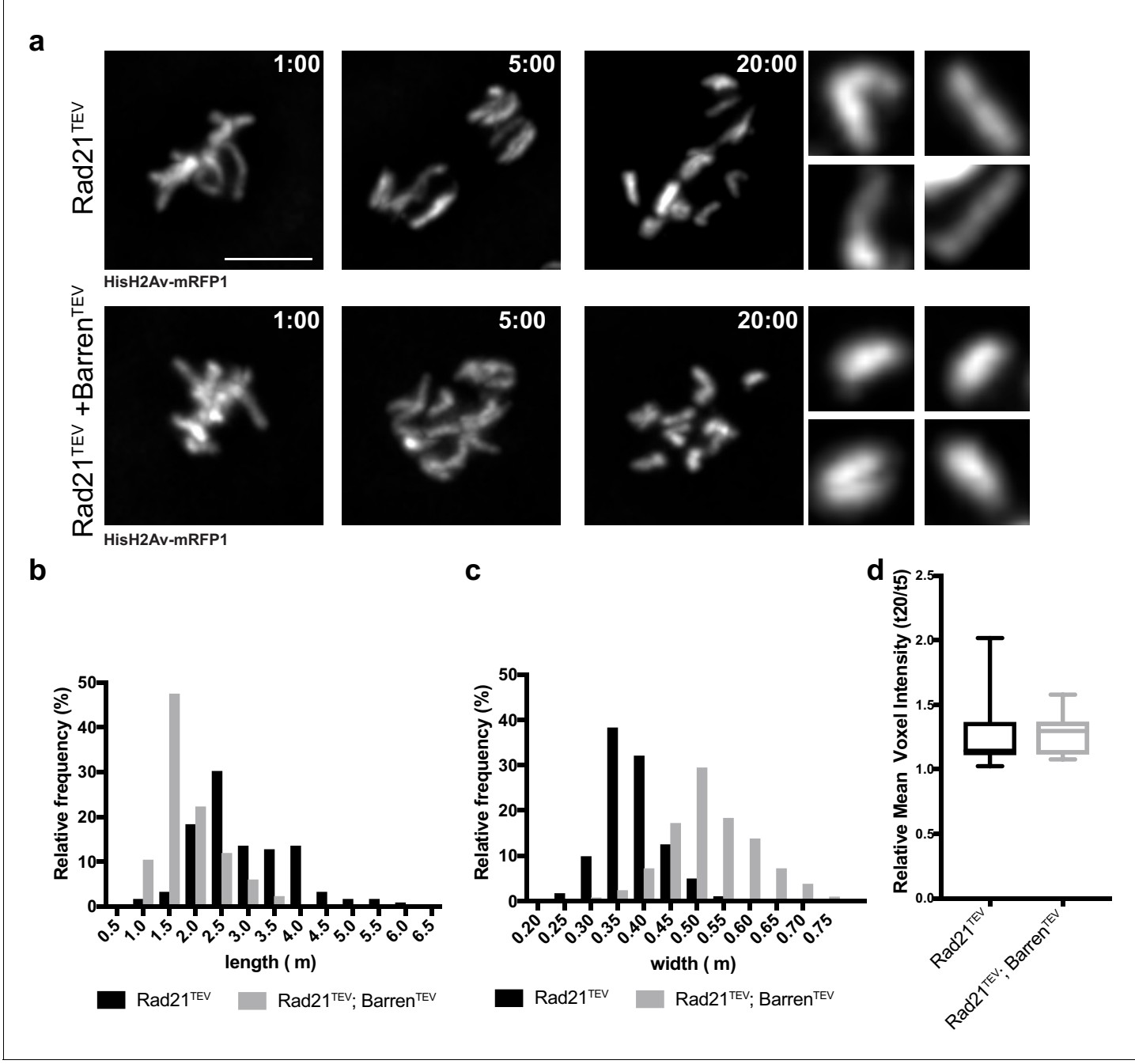

**Figure 5.** Chromosome over-compaction depends on sister-chromatid proximity. (**a**) Stills from metaphase-arrested embryos after injection of TEV protease in strains surviving solely on Rad21[TEV] (cohesin cleavage) or Rad21[TEV]+Barren[TEV] (cohesin and condensin cleavage); embryos also express HisH2B-RFP; scale bars, 5 µm. Insets show higher magnifications (3x) of single chromatids 20 min after TEV injection. Times (minutes:seconds) are relative to the time of TEV injection. (**b–c**) Relative frequency of sister chromatid length (**b**) and width (**c**) at 20 min after TEV injections (n ≥ 120 single chromatids from seven independent embryos for each experimental condition). (**d**) Mean voxel intensity of isolated single chromatids 20 min after TEV injections, normalized to mean voxel intensity 5 min past injection. (n ≥ 10 embryos for each experimental condition).

The following source data is available for figure 5:

**Source data 1.** Measurements of isolated chromatids upon cohesin/condensin inactivation.

inconsistencies in prior analysis on condensins depletion that as sample preparation, chromosome state, presence/absence of microtubules, or even the cell division type (mitosis vs meiosis) may play a major role in chromosome morphology. In contrast to condensin I inactivation, TopoII inhibition leads to rapid chromosome decompaction. These finding are consistent with the idea that metaphase chromosome structure is organized as a chromatin network resultant from self-entanglements of DNA strands, as initially proposed by biophysical studies on isolated chromosomes (*Kawamura et al., 2010*). Restricting/favouring chromosome entanglements may thus dictate the state of chromosome compaction.

Condensin has been previously proposed to interplay with TopoII, both for chromosome compaction and sister chromatid resolution. The exact details for this interaction, however, remain elusive. Both condensins and TopoII inactivation impair sister chromatid resolution (*Bhat et al., 1996*; *Clarke et al., 1993*; *Gerlich et al., 2006*; *Hagstrom et al., 2002*; *Hirano, 2006*; *Hudson et al., 2003*; *Oliveira et al., 2005*; *Ribeiro et al., 2009*; *Steffensen et al., 2001*; *Uemura et al., 1987*), suggesting these two molecules have cooperative roles on chromosome resolution. In contrast, cytological analyses suggest that condensin and TopoII have opposite roles in shaping mitotic chromatin (*Samejima et al., 2012*), raising further doubts on their functional interaction. It has long been hypothesized that condensin may impose directionality on TopoII reactions (*Baxter et al., 2011*; *Charbin et al., 2014*; *Coelho et al., 2003*; *Leonard et al., 2015*), as this enzyme is able to both decatenate and catenate DNA strands. But this model has been very difficult to formally prove. Studies in yeast using artificial circular mini-chromosomes, in which the levels of catenation can be directly measured, support that full decatenation by TopoII requires condensin activity (*Baxter and Aragón, 2012*; *Baxter et al., 2011*; *Charbin et al., 2014*). Whether the same is true in large and linear native chromosomes remained to be addressed, particularly as circular chromosomes are under different topological constrains when compared to linear ones. The experimental approach used in our study allowed the manipulation of native chromosomes, in their natural environment, providing evidence

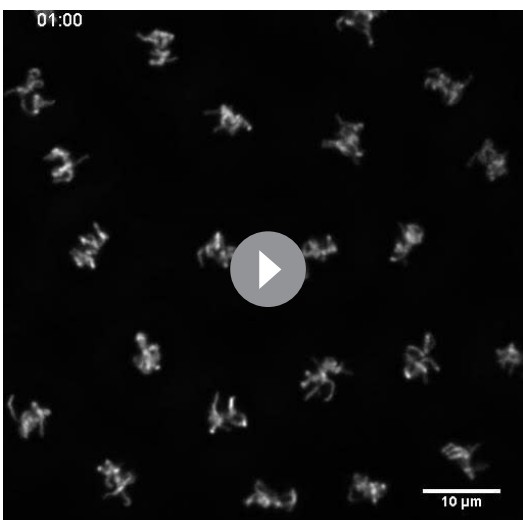

**Video 7.** Artificial induction of sister chromatid separation in metaphase-arrested embryos. Embryos expressing solely Rad21[TEV] and wild-type Barren were injected with 12 mg/ml of a dominant-negative form of the human E2 ubiquitin-conjugating enzyme (UbcH10[C114S]), to induce a metaphase arrest, and subsequently injected with 13 mg/ml TEV protease. Embryos also express His2B–RFP; scale bars, 10 μm. Times (minutes:seconds) are relative to the time of the second injection.

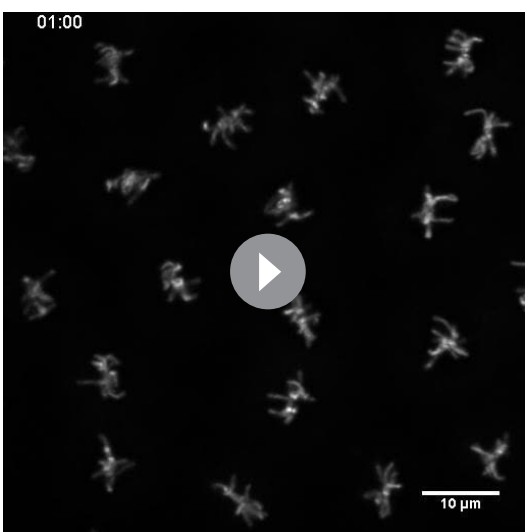

**Video 8.** Effect of condensin I inactivation on isolated sister chromatids. Embryos expressing uniquely TEV-sensitive Rad21 and Barren were injected with 12 mg/ml of a dominant-negative form of the human E2 ubiquitin-conjugating enzyme (UbcH10[C114S]), to induce a metaphase arrest, and subsequently injected with 13 mg/ml TEV protease. Embryos also express His2B–RFP; scale bars, 10 μm. Times (minutes:seconds) are relative to the time of the second injection.

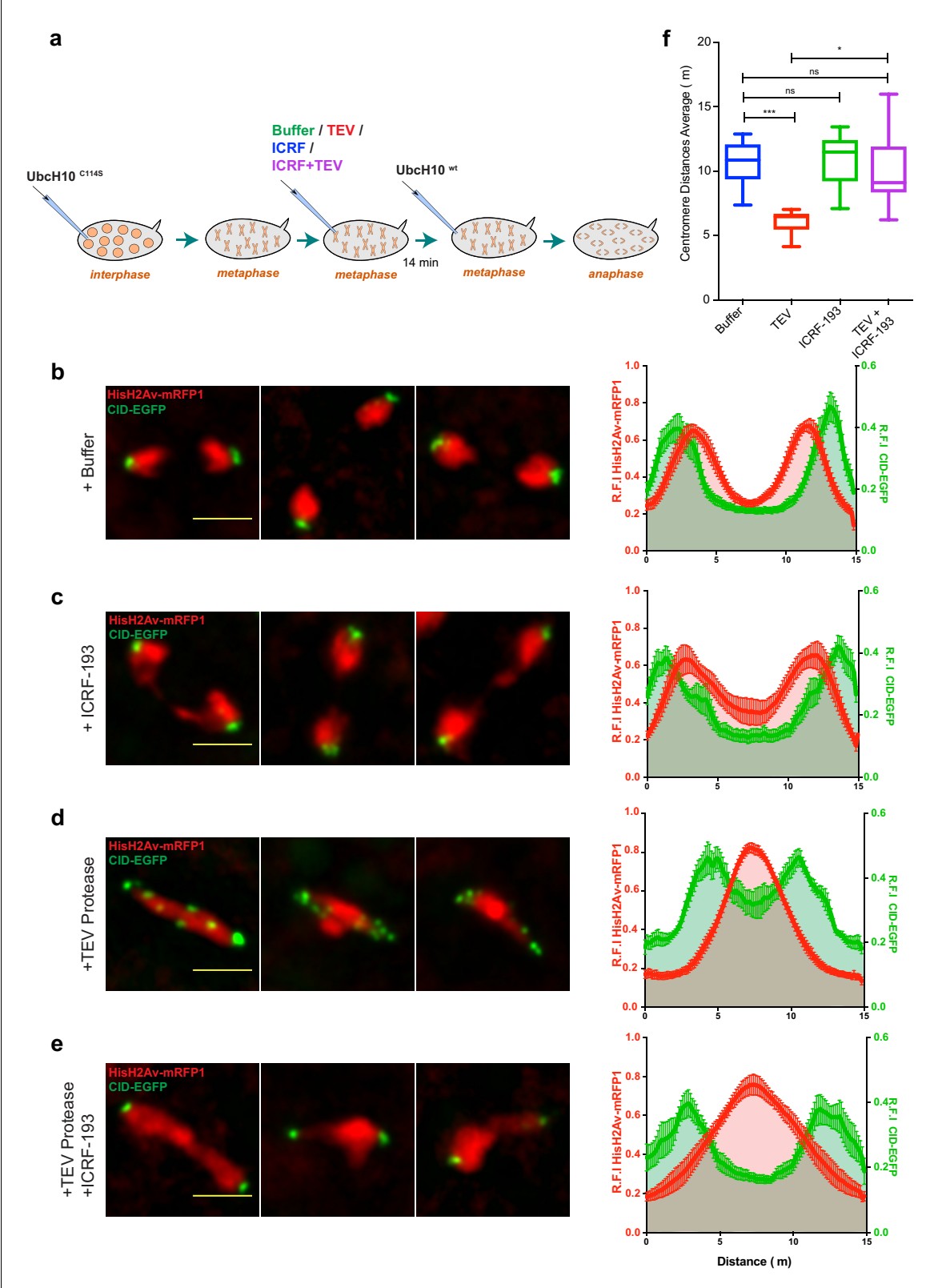

**Figure 6.** Condensin I inactivation results in TopoII-dependent sister chromatids intertwines and segregation failure. (**a**) Schematic representation of the experimental set-up. Embryos were arrested with 12 mg/ml UbcH10$^{C114S}$ and injected with buffer (**b**), 280 μM ICRF-193 (**c**), 13 mg/ml TEV protease (**d**) or TEV+ICRF-193, while in metaphase; Embryos were subsequently injected with 14 mg/ml of a wild-type version of UbcH10 to release them from the arrest. Images depict representative images of the anaphase; Graphs plot the relative distribution of HisH2Av-mRFP1 and Cid-EGFP across the 15 μm

*Figure 6 continued on next page*

*Figure 6 continued*

segregation plane, measured 4–6 min after anaphase onset. Graphs plot the average +_SEM of individual embryos (n ≥ 10 embryos for each experimental condition). For each embryo, at least eight anaphases were analysed. (f) Quantification of centromere distances during UbcH10[wt]-induced anaphase as in (b–e). Graphs plot the distances between segregating centromeres measured 6 min after anaphase onset (n ≥ 10 embryos for each experimental condition; for each embryo, at least eight anaphases were analysed). Statistical analysis was performed using the non-parametric Kruskal-Wallis test; ns p>0.05, *p<0.05; ***p<0.001.

The following source data and figure supplement are available for figure 6:

**Source data 1.** Measurements of segregation efficiency after metaphase-specific inactivation of condensin and/or Topoisomerase II.

**Figure supplement 1.** – Comparative analysis of segregation efficiency for condensin and/TopoII inhibition before mitosis (light colour) and during metaphase arrest/release (dark colour); Graphs plot the relative distribution of HisH2Av-mRFP1 (red) and Cid-EGFP (green) across a 20 µm segregation plane, measured 4–6 min after anaphase onset.

that upon removal of condensin I, previously separated sister chromatids re-intertwine in a TopoII-dependent manner. These findings are in agreement with a recent study that revealed that the resolution of sister chromatids from circular minichromosomes can be reverted by increased expression of TopoII (*Sen et al., 2016*). All together, these results support that condensin I is not directly necessary for TopoII catalytic activity, but rather to impose directionality on TopoII reactions, favouring resolution of the sister DNA molecules rather than their intertwine. Upon condensin I removal, creation of new links between previously separated DNA strands leads to their increased proximity, which may underlie the observed increase in chromosome compaction. Importantly, our studies reveal that

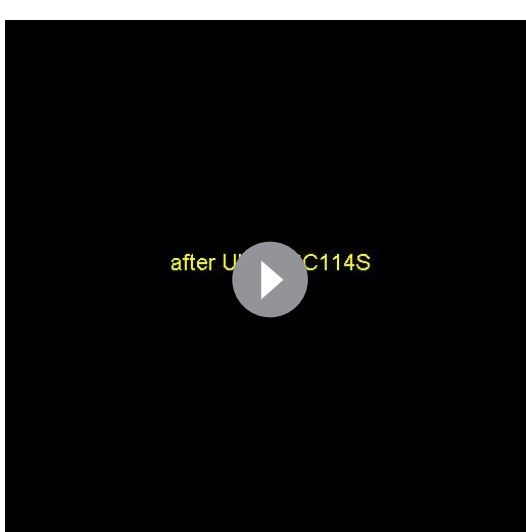

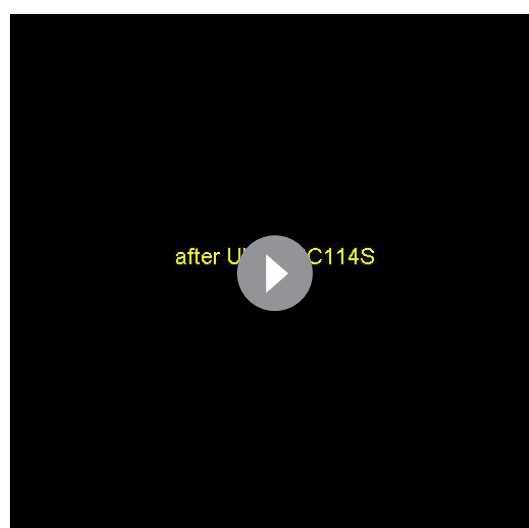

**Video 9.** Induced anaphase in control embryos. Embryos expressing solely Barren[TEV] were injected with 12 mg/ml of a dominant-negative form of the human E2 ubiquitin-conjugating enzyme (UbcH10[C114S]), to induce a metaphase arrest, and subsequently injected with buffer. After 14 min embryos were injected a wild-type version of UbcH10 to induce anaphase. Embryos also express His2A–mRFP1 (red) and Cid-EGFP (green); scale bars, 10 µm.

**Video 10.** Induced anaphase after timely inhibition of topoisomerase II. Embryos expressing solely Barren[TEV] were injected with 12 mg/ml of a dominant-negative form of the human E2 ubiquitin-conjugating enzyme (UbcH10[C114S]), to induce a metaphase arrest, and subsequently injected with 280 µM ICRF-193. After 14 min embryos were injected a wild-type version of UbcH10 to induce anaphase. Embryos also express His2A–mRFP1 (red) and Cid-EGFP (green); scale bars, 10 µm.

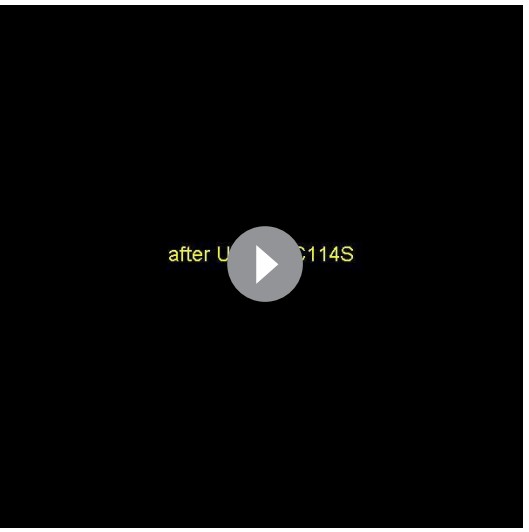

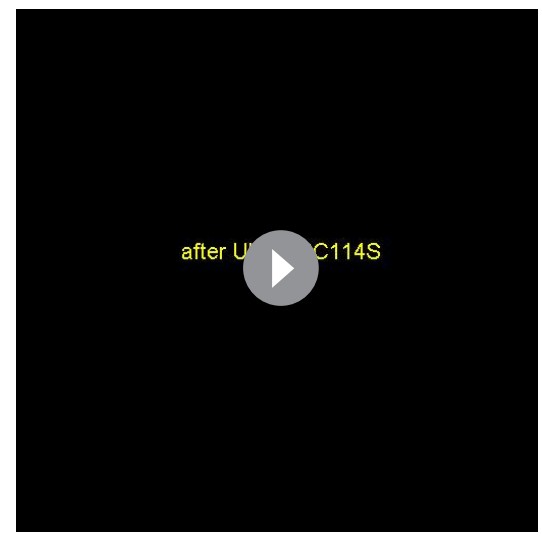

**Video 11.** Induced anaphase after timely inhibition of Condensin I. Embryos expressing solely Barren[TEV] were injected with 12 mg/ml of a dominant-negative form of the human E2 ubiquitin-conjugating enzyme (UbcH10[C114S]), to induce a metaphase arrest, and subsequently injected with 13 mg/ml TEV protease. After 14 min embryos were injected a wild-type version of UbcH10 to induce anaphase. Embryos also express His2A–mRFP1 (red) and Cid-EGFP (green); scale bars, 10 μm.

**Video 12.** Induced anaphase after timely inhibition of Condensin I and topoisomerase II. Embryos expressing solely Barren[TEV] were injected with 12 mg/ml of a dominant-negative form of the human E2 ubiquitin-conjugating enzyme (UbcH10[C114S]), to induce a metaphase arrest, and subsequently injected with a mix of 280 μM ICRF-193 and 13 mg/ml TEV protease. After 14 min embryos were injected a wild-type version of UbcH10 to induce anaphase. Embryos also express His2A–mRFP1 (red) and Cid-EGFP (green); scale bars, 10 μm.

**Table 1.** List of fly strains used in this study

| CHR#[*] | Genotype | Reference |
|---|---|---|
| 1418 | Barr[L305]/CyO | *Bhat et al. (1996)* (RRID:BDSC_4402) |
| 1421 | Df(2L)Exel7077/CyO | Blommington #7850 (RRID:BDSC_7850) |
| 1513 | w;; Barr(175 - 3TEV)-myc10 III.5 | This study |
| 1509 | w; Barr(175 - 3TEV)-myc10 II.1; | This study |
| 1522 | w;; Barr(389 - 3TEV)-myc10 III.2 | This study |
| 1514 | w;; Barr(437 - 3TEV)-myc10 III.1 | This study |
| 1520 | w;; Barr(600 - 3TEV)-myc10 III.3 | This study |
| 1525 | w;; Barr(wt)-myc10 III.1 | This study |
| 1560 | w; Barr[L305]/ Df(2L)Exel7077; Barr(175 - 3TEV)-myc10 III.5 | This study |
| 820 | w;; HisH2AvD-mRFP1 III.1, CGC (CID-EGFP) III.1 | *Schuh et al. (2007)* |
| 1564 | Df(2L)Exel7077 / CyO; HisH2AvD-mRFP1 III.1, CGC (CID-EGFP) III.1 | This study |
| | w; Barr[L305]/ Df(2L)Exel7077; Barr(175 - 3TEV)-myc10 III.5/ HisH2AvD-mRFP1 III.1, CGC (CID-EGFP) III.1 | |
| 629 | w;; Rad21[ex15], polyubiq-H2B-RFP, tubpr-Rad21(550-3TEV) -myc10 | *Oliveira et al. (2010)* |
| 1646 | w; Barr[L305], Barr(175 - 3TEV)-myc10 II.1; +/+ | This study |
| 1648 | w; Barr[L305], Barr(175 - 3TEV)-myc10 II.1; Rad21[ex15], polyubiq-H2B-RFP, tubpr-Rad21(550-3TEV) -myc10 | This study |

[*]Reference number in our internal lab fly database

TopoII is able to promote erroneous re-entanglements of sister chromatids throughout mitosis, an activity that needs to be constantly opposed by condensin I.

How condensin I is able to confer such directionality remains to be addressed. Condensins are enriched at the chromosome axis where they have been proposed to promote interactions within the same chromatid (*Ono et al., 2003*; *Steffensen et al., 2001*). Condensin I was shown to display significant turn-over on mitotic chromosomes (*Gerlich et al., 2006*; *Oliveira et al., 2007*) highlighting that its mode of action relies in dynamic reactions rather than statically holding chromatin loops. Bringing strands of DNA from the same chromatid in close proximity could alone favour sister chromatid decatenation by limiting the probability contacts between sister DNA molecules. Models that predict that DNA loops can extrude away from condensin have been hypothesized (*Goloborodko et al., 2016*; *Nasmyth, 2001*) and are better at explaining the directionally issue, as they provide a mechanism that inherently explains how condensins distinguish intra- versus inter-chromosomal looping. Random intrachromatid linkages are also possible (*Cheng et al., 2015*; *Cuylen et al., 2011*), although in this case additional mechanisms may ensure that connections in cis are favoured over linkages between sister- (and nearby) chromatids. Condensin I- mediated supercoiling of the DNA molecule has also been proposed to change DNA structure to favour DNA decatenation activity (*Baxter and Aragón, 2012*; *Baxter et al., 2011*; *Sen et al., 2016*), although it is yet to be determined whether the supercoiling activity of this complex can account for all the phenotypes associated with condensin loss.

Our analysis further reveals that maintenance of chromosome architecture, particularly sister chromatid resolution, is not a unidirectional process but instead a much more dynamic reaction than previously anticipated. It is conceivable that the highly compacted chromatin state present in metaphase chromosomes could, on its own, shift TopoII reaction towards sister chromatid re-entanglement given the increased proximity between DNA strands. Condensin I would thus counteract an inherent tendency of chromosomes to re-intertwine, a reaction necessary throughout metaphase. Additionally, it is possible that a dynamic balance of chromosome entanglements allows remodelling of chromosome architecture, providing chromosomes with plasticity to counteract the cytoplasmic drag faced during dynamic movements. Energy released during these reactions could potentially be used to further facilitate chromosome movement. Mitotic chromosomes should thus be visualized as highly dynamic structures during mitosis, whose re-shaping may be fundamental for the fidelity of their own segregation.

## Materials and methods

### Fly strains

To destroy condensin by TEV protease-mediated cleavage, strains carrying solely TEV-sensitive Barren versions were produced. A construct carrying a ~4.7 kb Barren genomic region was used as a starting point (kindly provided by Beat Suter, Institute of Cell Biology, University of Bern). This region contains the regulatory sequences and was previously shown to restore Barren function (*Masrouha et al., 2003*). This construct was engineered to add a 10xMyc sequence at the C-terminus of Barren. Three consecutive TEV recognition sites were placed at different positions (corresponding to a.a. 175, a.a. 389, a.a. 437 and a.a 600). Cloning details are available upon request. Each variant of genomic Barren with different TEV sites was cloned into pCaSpeR4 vector used for fly transformation. Transgenic flies were produced by P-element integration (BestGene Inc, Chino Hills, CA). Transgenes were placed in a $Barr^{L305}$ background, a Barren null allele (*Bhat et al., 1996*), over a deficiency for the corresponding genomic region (*Df(2L)Exel7077*, stock #7850 from Bloomington stock center). To destroy cohesin by TEV-protease we used strains carrying $Rad21^{TEV}$, previously described (*Oliveira et al., 2010*; *Pauli et al., 2008*). Fly strains also expressed His2AvD–mRFP1 or polyubiquitin His2B–RFP, to monitor DNA and EGFP–Cid to monitor centromeres (*Schuh et al., 2007*). A list with detailed genotypes can be found in *Table 1*.

### Microinjections

Microinjection experiments were performed as previously described (*Oliveira et al., 2010*). 1–1.5 hr old embryos (or 0–30 min for mRNA injections) were collected and processed according to standard protocols, and embryos were injected at the posterior pole (up to three sequential injections) using

a Burleigh Thorlabs Micromanipulator, a Femtojet microinjection system (Eppendorf, Germany), and pre-pulled Femtotip I needles (Eppendorf). Embryos were injected with buffer, drugs or proteins purified from *E. coli* at the following concentrations: Buffer (20 mM Tris-HCl at pH 8.0, 1 mM EDTA, 50 mM NaCl and 2 mM DTT), 13 mg/ml TEV protease in TEV buffer, 12 mg/ml UbcH10$^{C114S}$, 14 mg/ml UbcH10$^{wt}$ and/or 280 µM ICRF-193 (Sigma-Aldrich, St Louis, MO).

## Protein purification

Purified TEV protease was described previously (*Haering et al., 2008*). Purification of UbcH10$^{wt}$ and UbcH10$^{C114S}$ was performed from BL21 cells as previously described (*Oliveira et al., 2010*), with minor modifications, as follows. Bacterial cells were grown for 16 hr at 37°C, 225 rpm. This pre-culture was used to inoculate fresh LB media and cells were allowed to grow until 0.8/1 ODs. Cultures were then induced with 1 mM IPTG and after 4 hr of induction at 37°C, 225 rpm, cells were harvested. Pellets were ressuspended in Lysis Buffer (20 mM Tris-HCL pH7.5, 0.5M NaCl, 5 mM Imidazole with protease inhibitors) and sonicated 5x on ice in 30 s cycles (power 5- Sonicator XL2020, Misonix, Farmingdale, NY). The soluble fraction of the extracts was then incubated in TALON Metal Affinity Resin (Takara Bio Inc. , Japan), according to manufacturer's instructions. After several washes with Lysis Buffer, the resin coated with the protein was packed into a Poly-Prep Chromatography Column (Biorad, Hercules, CA). Proteins were eluted in the same buffer with 300 mM imidazole. For buffer exchange, purified UbcH10$^{wt}$ and UbcH10$^{C114S}$ proteins were dialyzed overnight, at 4°C, in a Slide-a-Lyzer 7 KDa Dialysis cassettes (Thermo Scientific, Waltham, MA). Final storage buffer was 20 mM Tris-HCL pH7.5, 0.3M NaCl. The purified proteins were concentrated in a Vivaspin 6 Centrifugal Concentrator MWCO 10.000 KDa (GE Healthcare, Issaquah, WA).

## mRNA synthesis

Barren$^{175TEV}$-EGFP was cloned into a pRNA plasmid and mRNA was synthesized by in vitro transcription with the mMessage mMachine T3 kit (Ambion, Austin, TX), followed by purification with RNeasy kit (Qiagen, Germany), and elution in RNase-free water. To probe for the efficiency of Barren$^{TEV}$ removal (*Figure 1C*), 0–30 min old embryos surviving only on Barren$^{TEV}$-Myc were injected with Barren$^{TEV}$-EGFP mRNA in pure water at ~2.2 µg/µl. Embryos were left to develop at 22°C for 1,5–2 hr, to allow for protein translation, before the subsequent injections.

## In vitro cleavage experiments

Ovaries were dissected from females and homogenized in PBS. Extracts were sonicated for 2 min in a water-bath (power 5- Sonicator XL2020, Misonix). After centrifugation for 10 min at 15.000 rpm at 4°C, the supernatant was removed and adjusted to a final concentration of 2 µg/µl. For cleavage experiments, 80 µl of extract were incubated with 2 µg of TEV protease. At the indicated time points, 10 µl of the reaction were diluted with sample buffer, boiled and stored at −20°C.

## Western-blot

Samples were loaded on a 10% SDS-gel for electrophoresis and transferred onto a membrane (Immun-Blot PVDF, Biorad). Western-blot analysis was performed according to standard protocols using the following antibodies: anti myc-tag (1:200, Santa Cruz Biotechnology, Dallas, TX, Cat# sc-47694 RRID:AB_627266), anti-α-tubulin (1:50.000, DM1A, Sigma-Aldrich Cat# T9026 RRID:AB_477593) and anti-Barren (1:3000, kindly provided by Hugo Bellen, (*Bhat et al., 1996*), RRID:AB_2567044).

## Microscopy

Aligned embryos on coverslips were covered with Series 700 halocarbon oil (Sigma-Aldrich). Imaging of embryos after mRNA injection (*Figure 1c*) was performed with a spinning disc Revolution XD microscope (Andor, UK) at 22°C. Stacks of around 20 frames 1 µm were taken at indicated times using a 100 × 1.4 oil immersion objective (Nikon, Japan) and iXon +512 EMCCD camera (Andor). Time-lapse microscopy was performed with an inverted wide-field DeltaVision microscope (Applied Precision Inc., Issaquah, WA) at 18–20°C in a temperature-controlled room. One stack of ~20 frames (0.8 µm apart) was acquired every 1 or 2 min using a 100 × 1.4 oil immersion objective (Olympus, Japan) and an EMCCD camera (Roper Cascade 1024, Roper Technologies,

Inc., Sarasota, FL). Widefield images were restored by deconvolution with the Huygens v15.10/16.10 deconvolution software using a calculated point-spread function (RRID:SCR_014237, Scientific Volume Imaging, The Netherlands). Movies were assembled using FIJI software (RRID:SCR_002285) (*Schindelin et al., 2012*) and selected stills were processed with Photoshop CS6 (Adobe Systems Incorporated, San Jose, CA).

### Quantitative imaging analysis

For the quantification of chromosome condensation presented in *Figure 3g* and *Figure 3—figure supplement 1*, deconvolved images were analyzed using Imaris v6.1 software (RRID:SCR_007370, Bitplane, Switzerland). The same metaphase was tracked over time and average values for mean voxel intensity, volume and surface area were normalized to the first frame after injection. For the fluorescence profiles presented in *Figures 3f* and *6b–e,a* wide 15 µm-long line was placed manually along the segregation plane and measured using the 'Plot Profile' function on FIJI software. For each data set, values were normalized to the maximum. Measurements of single chromatids width and length were performed on projected images (maximum intensity projection), using FIJI software and single chromatids mean voxel intensity measurements were performed using Imaris software. Quantification of chromosome movement (*Figure 4*) was performed as previously described (*Mirkovic et al., 2015*). Briefly, HisH2B-RFP was imaged at 1 min intervals. Images were segmented to select the chromosomal regions, based on an automatic threshold (set in the first frame after TEV injection), to create binary images. For each movie, a walking average of 3 frames was produced (using kymograph plug-in, written by J. Rietdorf and A. Seitz, EMBL, Heidelberg, Germany) creating a merged image in which the intensity is proportional to the overlap between consecutive frames. Intensity profiles were used to estimate the percentage of non-overlapping, 2- frame overlap and 3-frame overlap pixels. Graphic representation was performed using Prism seven software (RRID:SCR_002798, GraphPad, La Jolla, CA).

### Statistical analysis

To compare the average of the centromere distances between each experimental condition (*Figure 6f*), at least 10 independent embryos were analyzed. Statistical analysis was performed using Prism seven software (RRID:SCR_002798). Given that some datasets did not pass the normality test (D'Agostino and Pearson normality test), multiple comparisons were performed using the non-parametric Krustal-Wallis test.

## Acknowledgements

We thank S Heidmann, H Bellen and B Suter for fly strains, antibodies and plasmids, I Telley, A Athanasiadis for helpful advice, the Advance Imaging Unit and fly facility for technical assistance, all the members of the RAO laboratory for discussions and comments, and Kim Nasmyth for comments on the manuscript. EP holds a PhD fellowship from the Fundação para a Ciência e Tecnologia (SRFH/BD/52172/2013). This work was supported by the following grants awarded to RAO: FCT Investigator grant (IF/00851/2012/CP0185/CT0004 WP1), Marie Curie Career Integration Grant (MCCIG321883/CCC), EMBO Installation Grant (IG2778) and European Research Council Starting Grant (ERC-2014-STG-638917).

## Additional information

### Funding

| Funder | Grant reference number | Author |
| --- | --- | --- |
| Fundação para a Ciência e a Tecnologia | SRFH/BD/52172/2013 | Ewa Piskadlo |
| European Commission | MCCIG321883/CCC | Raquel A Oliveira |
| European Molecular Biology Organization | IG2778 | Raquel A Oliveira |
| Fundação para a Ciência e a | IF/0085½012/CP0185/ | Raquel A Oliveira |

| Tecnologia | CT0004 WP1 | |
| European Commission | ERC-2014-STG-638917-ChromoCellDev | Raquel A Oliveira |

The funders had no role in study design, data collection and interpretation, or the decision to submit the work for publication.

### Author contributions

EP, Conceptualization, Resources, Formal analysis, Investigation, Methodology, Writing—review and editing; AT, Resources, Methodology, Writing—review and editing, Protein purification; RAO, Conceptualization, Formal analysis, Supervision, Funding acquisition, Writing—original draft

### Author ORCIDs

Ewa Piskadlo, http://orcid.org/0000-0003-0857-6744
Raquel A Oliveira, http://orcid.org/0000-0002-8293-8603

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
