## [Decision Letter]

Thank you for submitting your article "Metaphase chromosome structure is dynamically maintained by Condensin I-directed DNA (de)catenation" for consideration by *eLife*. Your article has been favorably evaluated by Anna Akhmanova (Senior Editor) and three reviewers, one of whom is a member of our Board of Reviewing Editors. The following individual involved in review of your submission has agreed to reveal their identity: Luis Aragon (Reviewer #2).

The reviewers have discussed the reviews with one another and the Reviewing Editor has drafted this decision to help you prepare a revised submission.

Three reviewers have assessed your manuscript, and the consensus opinion is that the experimental system is powerful; that the technical execution of the experiments was outstanding; and that the material is suitable for publication after revisions have been undertaken. Below, the overall view, comments and concerns are listed.

A number of studies have demonstrated the importance of condensin complexes for the structural organization and segregation of mitotic chromosomes, but the precise molecular function of these protein complexes has remained incompletely understood. Oliveira and colleagues report the effects of inactivating condensin I by TEV protease-mediated cleavage of the kleisin subunit at metaphase in *Drosophila* embryos. They observe that sister chromatids fail to resolve while centromeres stretch to the cell poles upon anaphase onset. Interestingly, the density of chromatin mass simultaneously increases in a Topo II-dependent manner, which suggests that sister chromatids that had been previously separated re-entangle in the absence of condensin function. This conclusion is supported by the facts that (2) the density of already separated sister chromatids does not increase upon condensin cleavage and (1) simultaneous inactivation of Topo II decreases the segregation defect caused by condensin cleavage. The authors conclude that condensin's role is to maintain sister separation beyond the initial condensation process by preventing re-catenation by Topo II.

Specific comments:

A) The authors miss an opportunity to make their discussion and interpretation of their data as incisive and authoritative as it could be. They need to make it clear that the interpretation that hyper-compaction of chromosome arms is a consequence of increased catenation is a hypothesis; it is not directly demonstrated, only inferred. Furthermore, central to the biological relevance of the observations made is a precise definition of the term 'condensation'. The authors here have a chance to clearly define 'condensation' [see below] and compare it with 'compaction' or 'over-compaction' which is the phenotype measured in the imaging assays; and which does not necessarily reflect a specific 'structure', determined by a specific mechanism. Does the intermingling of sister chromatids, which follows condensin inactivation, indeed reflect the same biological process that takes place during the folding of mitotic chromosomes during mitotic prophase? Presumably not. The statement that condensin removal leads to 'over-condensation' of chromatids is terribly misleading if all that happens is that chromatids get intertwined ['…leads to 'over-compaction as a consequence of chromosome intertwining' would work]. Condensation [the formation of an ordered (intra-chromatid) architecture that cannot just be achieved by compacting chromatin fibers in a random manner] is actually precisely the opposite of what is being observed in this case. The conclusions that "Topo II may alone drive chromosome condensation" is hence a conceptual misinterpretation, as is would suggest that TopoII would either actively drive intra-chromatid catenation of that condensation is a self-organizing principle of mitotic chromatin and TopoII facilitated this process. In either case, only limited conclusions about the principles of the formation of mitotic chromosomes during prophase (i.e. the condensation mechanism) can be drawn from the intertwining of metaphase sister chromatids that results from condensin inactivation.

B) It would be interesting to know whether when TopoII is depleted in metaphases arrested by SAC, if segregation is more defective than when TopoII is inactivated in metaphases obtained by UbcH10(C114S) (where SAC is not engaged). Yeast work (Farcas et al. 2011; Sen et al. 2016) has shown that plasmids contain higher levels of catenanes in nocodazole arrests than cdc20-depletion arrests, and that condensin activity is different in both arrest. Is this also the case here? The authors should comment on whether this experiment is technically feasible, whether they have this information, and whether they would consider it desirable to include this information and the result.

C) Could one compare the levels of sister chromatid (re-)entangling after condensin inactivation in metaphase to the levels of intertwining before mitotic chromosomes start to form (i.e. after S phase). Perhaps the authors could measure histone H2Av-mRFP1 densities after TopoII inhibition in G2 phase cells and compare these values to the values measured after condensin activation during metaphase (Figure 3)? The authors might consider whether this would be a 'doable' and informative experiment.

D) Does TEV cleavage remove Barren also from centromeres? This is difficult to judge, since the CID-EGFP signal masks the Barren-TEV-EGFP signal in Figure 1. It might be useful to show images without a labelled version of CID [it would also be useful if the authors defined 'CID' at its first use].

---

## [Author Response]

Specific comments:

A) The authors miss an opportunity to make their discussion and interpretation of their data as incisive and authoritative as it could be. They need to make it clear that the interpretation that hyper-compaction of chromosome arms is a consequence of increased catenation is a hypothesis; it is not directly demonstrated, only inferred.

We have changed the text accordingly throughout the manuscript to present the results in a more direct and incisive manner. We additionally removed the consequential tone between the chromosome compaction observations and chromatid re-entanglement, and present the link between the two as a possibility. Although these two observations are likely to be connected, we agree it is not fully demonstrated. The Abstract was therefore considerably modified to highlight the major findings of the paper (chromatid-re-intertwining).

Furthermore, central to the biological relevance of the observations made is a precise definition of the term 'condensation'. The authors here have a chance to clearly define 'condensation' [see below] and compare it with 'compaction' or 'over-compaction' which is the phenotype measured in the imaging assays; and which does not necessarily reflect a specific 'structure', determined by a specific mechanism. Does the intermingling of sister chromatids, which follows condensin inactivation, indeed reflect the same biological process that takes place during the folding of mitotic chromosomes during mitotic prophase? Presumably not. The statement that condensin removal leads to 'over-condensation' of chromatids is terribly misleading if all that happens is that chromatids get intertwined ['…leads to 'over-compaction as a consequence of chromosome intertwining' would work]. Condensation [the formation of an ordered (intra-chromatid) architecture that cannot just be achieved by compacting chromatin fibers in a random manner] is actually precisely the opposite of what is being observed in this case. The conclusions that "Topo II may alone drive chromosome condensation" is hence a conceptual misinterpretation, as is would suggest that TopoII would either actively drive intra-chromatid catenation of that condensation is a self-organizing principle of mitotic chromatin and TopoII facilitated this process. In either case, only limited conclusions about the principles of the formation of mitotic chromosomes during prophase (i.e. the condensation mechanism) can be drawn from the intertwining of metaphase sister chromatids that results from condensin inactivation.

We thank the reviewers for pointing out this very important point. Indeed our experiments reveal mechanisms required for *maintenance* of chromosome architecture, which may or may not be similar to the process of chromosome condensation. We have changed the manuscript so that the term condensation is only used to refer to the normal process of mitotic chromosome assembly and used solely the term compaction to refer to the chromatin density we measure upon metaphase perturbations. We also defined specifically the term compaction. You can now read: “We defined chromosome compaction by degree of chromatin density, inferred from the signal of fluorescently labelled histone H2Av-mRFP1.”

B) It would be interesting to know whether when TopoII is depleted in metaphases arrested by SAC, if segregation is more defective than when TopoII is inactivated in metaphases obtained by UbcH10(C114S) (where SAC is not engaged). Yeast work (Farcas et al. 2011; Sen et al. 2016) has shown that plasmids contain higher levels of catenanes in nocodazole arrests than cdc20-depletion arrests, and that condensin activity is different in both arrest. Is this also the case here? The authors should comment on whether this experiment is technically feasible, whether they have this information, and whether they would consider it desirable to include this information and the result.

We agree that it would be very interesting to address the efficiency of DNA decatenation in a SAC-arrest situation, as the absence of spindle forces is likely to play a major role in this process. We have thus attempted to address this important issue by several means. However, several experimental limitations precluded a definitive answer. Below is a summary of our attempts to address the issue and the reasons why we considered these experiments inconclusive. A major technical limitation is the inability to measure the levels of catenation specifically in metaphase (a clear advantage of the referred studies in budding yeast, by the use of purifiable minichromosomes). Throughout the manuscript we used an alternative method, the efficiency of chromosome segregation during anaphase, as an indirect read-out for the amount of catenation. It is thus virtually impossible to access the catenation levels by this method in the absence of a functional spindle, as no anaphase takes place. We therefore opted for not including them in the manuscript.

1) We have attempted to drive cells into anaphase upon SAC-arrest imposed by spindle depolymerization. This was achieved by colchicine arrest to depolymerize microtubules followed by a Mad2deltaC injection, which works as a dominant negative for the SAC arrest. This method was efficient in driving cells out of mitosis. However, in these experiments, chromosomes simply exited mitosis without any evident separation (formed a tetraploid nucleus), and it was thus impossible to estimate catenation levels under these conditions.

2) We additionally attempted to use centromere separation in metaphase as an alternative measure of the amount of catenation. TEV-mediated cleavage of cohesin in embryos previously injected with colchicine resulted in a ~ 30% increase in the distances between sister-centromeres without full separation. The absence of full separation of sister chromatids could potentially be interpreted as a read-out of residual catenation, which would be sufficient to keep sister DNA molecules in proximity. However, ongoing studies in the lab revealed that chromatids that have been previously fully separated have a tendency to stay together upon disruption of the mitotic spindle. We therefore trust that the proximity between sister-chromatids observed upon cohesin cleavage is not a reliable read-out of the amount of DNA catenation as chromatids tend to “stick to each other” even when fully separated.

If the reviewers have any suggestions for an alternative experimental setup that could conclusively address this point we would be happy to consider it.

C) Could one compare the levels of sister chromatid (re-)entangling after condensin inactivation in metaphase to the levels of intertwining before mitotic chromosomes start to form (i.e. after S phase). Perhaps the authors could measure histone H2Av-mRFP1 densities after TopoII inhibition in G2 phase cells and compare these values to the values measured after condensin activation during metaphase (Figure 3)? The authors might consider whether this would be a 'doable' and informative experiment.

We have included this comparative analysis as a supplement to Figure 6. As expected, Topo II inhibition before mitotic entry leads to thick chromatin bridges. By contrast, inactivation in metaphase chromosomes produces a much milder phenotype, reinforcing that the degree of catenation in metaphase chromosomes is indeed residual. It was also interesting to note that the defects associated with condensin inactivation before mitosis were, by contrast, milder than observed upon condensin inactivation in metaphase-arrested chromosomes. This further supports the notion that TopoII dependent re-entanglement should be actively suppressed during metaphase, as the degree of re-entanglement in a prolonged metaphase may be even higher than the starting point.

D) Does TEV cleavage remove Barren also from centromeres? This is difficult to judge, since the CID-EGFP signal masks the Barren-TEV-EGFP signal in Figure 1. It might be useful to show images without a labelled version of CID [it would also be useful if the authors defined 'CID' at its first use].

To address this point, we performed similar cleavage experiments in strains that do not express CID-GFP maker. This enables tracing Barren^TEV^-EGFP alone, confirming that after TEV injection Barren is rapidly removed from all chromosome regions, including centromeres. We replaced the previous figure with this new result (Figure 1). We have also included a description of CID, as suggested (see subsection “A TEV-protease mediated system to inactivate condensin I in *Drosophila melanogaster*”).